



# Supershear crack propagation in snow slab avalanche release: new insights from numerical simulations and field measurements

Grégoire Bobillier[1,2,†], Bertil Trottet[3,†], Bastian Bergfeld[1], Ron Simenhois[4], Alec van Herwijnen[1], Jürg Schweizer[1], Johan Gaume[1,2,5]

[1] WSL Institute for Snow and Avalanche Research SLF, Davos, Switzerland.
[2] Climate Change, Extremes, and Natural Hazards in Alpine Regions Research Center CERC, Davos, Switzerland.
[3] École Polytechnique Fédérale de Lausanne, Switzerland.
[4] Colorado Avalanche Information Center, Boulder, CO, USA.
[5] Institute for Geotechnical Engineering, ETH Zurich, Switzerland.
[†] Equal contribution

*Correspondence to* Grégoire Bobillier (gregoire.bobillier@slf.ch)

**Abstract.** The release process of dry-snow slab avalanches begins with a localized failure within a porous, weak snow layer that lies beneath a cohesive slab. Subsequently, rapid crack propagation may occur within the weak layer, eventually leading to a tensile fracture across the slab, resulting, if the slope is steep enough, to its detachment and sliding. The dynamics of crack propagation is believed to influence the size of the release area. However, the relationship between crack propagation dynamics and avalanche size remains incompletely understood. Notably, crack propagation speeds estimated from avalanche video analysis are almost one order of magnitude larger than speeds typically measured in field experiments. To shed more light on this discrepancy and avalanche release processes, we used discrete (DEM: discrete element method) and continuum (MPM: material point method) numerical methods to simulate the so-called propagation saw test (PST). On low angle terrain, our models showed that the weak layer failed mainly due to a compressive stress peak at the crack tip induced by weak layer collapse and the resulting slab bending. On steep slopes, we observed the emergence of a supershear crack propagation regime: the crack speed becomes higher than the slab shear wave speed. This transition occurs if the crack propagates over a distance larger than the super-critical crack length (approximately 5 m). Above the super-critical crack length, the fracture is mainly driven by the slope-parallel gravitational pull of the slab (tension) and, thus, shear stresses in the weak layer. These findings represent an essential additional piece in the dry-snow slab avalanche formation puzzle.

## 1. Introduction

Forecasting the avalanche danger is vital in snow-covered mountain regions and relies on a solid comprehension of avalanche release processes. Among the various types of avalanches, accidents mainly arise from dry-snow slab avalanches. Dry-snow slab avalanches release is a multi-scale process that requires the presence of a highly porous weak snow layer buried beneath a cohesive snow slab. It starts with the formation of a localized failure within the weak layer induced by a perturbation, e.g., an additional load such as a skier. A crack may then form that rapidly propagates within the weak layer across the slope. During





the propagation process, the slab can eventually fracture and detach if the slope angle is greater than the friction angle of the crushed weak layer (approx. 30°; Mcclung, 1979; Schweizer et al., 2003; Van Herwijnen and Heierli, 2009). During the last two decades, our understanding of slab avalanche formation processes has greatly improved, in particular through the
development of a fracture mechanical field test: the propagation saw test (PST; Gauthier and Jamieson, 2006; Sigrist and Schweizer, 2007; Van Herwijnen and Jamieson, 2005). The PST consists of an isolated snow column containing a pre-identified weak layer. A crack is manually initiated in the weak layer with a snow saw until a critical crack length is reached, after which crack propagation is self-sustained without additional loading. Using a high-resolution and high-speed camera to capture the PST side wall deformation enables us to analyze the mechanical behavior of the different snowpack layers, such
as slab deformation and weak layer structural collapse during crack initiation and dynamic propagation. Furthermore, several analytical and numerical PST models based on fracture and/or continuum mechanics have been developed to investigate crack propagation behavior and provide additional insights into the mechanics involved (Benedetti et al., 2019; Chiaia et al., 2008; Gaume et al., 2013; Gaume et al., 2015; Gaume et al., 2017; Gaume et al., 2018; Rosendahl and Weissgraeber, 2020a, b).

The dynamics of crack propagation are believed to influence the extent of the release area and, consequently, the avalanche
size. Recognizing this, the PST has been increasingly used to investigate and explore the complex driving mechanisms governing the crack propagation dynamics (Bergfeld et al., 2022; Bergfeld et al., 2023; Bobillier et al., 2021; Gaume et al., 2018; Trottet et al., 2022; Bobillier et al., 2024). In recent numerical studies by Bobillier et al. (2021) and Trottet et al. (2022) emergence of a steady-state crack propagation regime was highlighted using discrete (discrete element method, DEM) and continuum (material point method, MPM) numerical methods, respectively. On flat terrain, the weak layer fails mostly due to
a compressive stress peak induced by the bending of the slab, which results from the structural collapse of the weak layer behind the crack tip. Consequently, the speed of crack propagation is bounded by the corresponding slab Rayleigh wave speed. On steep slopes, a supershear crack propagation regime may emerge; the crack speed becomes higher than the slab shear wave speed. This transition occurs if the crack propagates over a distance larger than the so-called super-critical crack length (Approximately 5 m; Trottet et al., 2022). Beyond this length, the fracture is mainly driven by the slope-parallel gravitational
pull of the slab (tension) and, thus, shear stresses in the weak layer (Bobillier et al., 2024).

We aim to describe the processes involved during the two steady-state regimes of weak layer crack propagation: sub-Rayleigh and supershear. To this end, we performed numerical simulations (using continuum and discrete numerical methods) and analyzed field measurements. We studied weak layer crack propagation dynamics during propagation saw tests (PST) and full-scale avalanches using high-speed videos of PST experiments and a high-quality avalanche movie. To simulate propagation
saw tests, we employed both the discrete element and the material point method, and compared the simulation results.




## 2. Data and methods

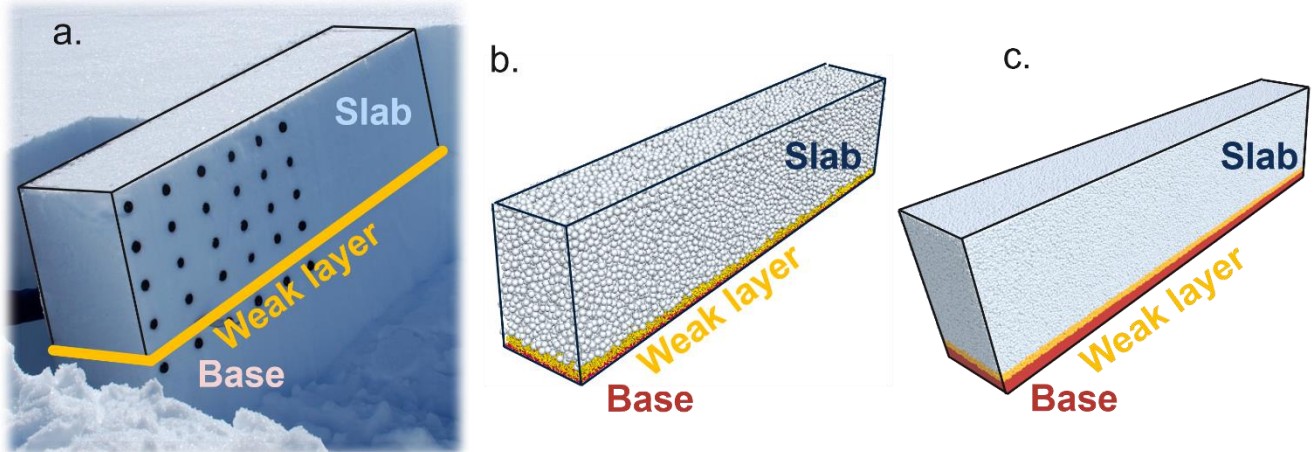

**Figure 1: (a) Experimental Propagation Saw Test (PST; photo: J. Schweizer). The black dots are markers used to measure slab deformation (particle tracking). (b) Discrete Element Method PST simulation, length 2 meters. (c) Material Point Method PST simulation.**

**Propagation saw test (PST)**

In the Propagation Saw Test (PST), a snow column consisting of a slab, a weak layer and a base layer is isolated (Figure 1). A manual cut is made in the weak layer with a saw until the critical length is reached ($a_c$). Once this critical length is reached, crack propagation becomes self-sustained without requiring additional loading or manually expanding the crack. Over the last decades, the PST has been used to better understand the mechanisms driving the onset and the crack propagation dynamics (e.g., Van Herwijnen et al., 2010). Particle tracking velocimetry (PTV) and/or digital image correlation (DIC) techniques were applied to measure slab deformation and derive crack propagation speed. Recently, this fracture mechanical field test has been extended to study dynamic crack propagation behavior over long distances (approximately 10 m; Bergfeld, 2022). Moreover, given the inherent complexities and uncertainties in snow mechanics, the slab component within the PST setup is commonly represented as a homogeneous layer with simplified parameters such as effective elastic modulus. For instance, in our study, we approximated the effective elastic modulus of the slab as a function of density as suggested by Scapozza (2004). This approximation was crucial for estimating the speed of elastic waves propagating within the snow medium, a fundamental aspect in understanding the behavior observed during field experiments.

**Discrete element method (DEM)**

We used the 3D DEM model that was developed to investigate crack propagation behavior in a PST experiment (Bobillier et al., 2021). Initially introduced by Cundall and Strack (1979), DEM is a numerical tool including numerous discrete interacting particles, commonly used to study large deformations in granular-like assemblies. Here, we use the commercial software





PFC3D (v5) developed by Itasca. The simulated PSTs consist of three layers: a rigid basal layer, a transversely isotropic weak layer, similar to layers of surface hoar or facets, and a dense and uniform slab layer. The weak layer was created using cohesive

ballistic deposition, resulting in a porosity of 80 % and a thickness ($h$) of 0.02 m. The slab layer was generated by cohesionless ballistic deposition with a porosity of 45 % and a thickness ($H$) of 0.5 m. We employed the parallel-bond contact model (PBM) for each layer to model particle interactions. The PBM component acts in parallel with a classical linear contact model and establishes an elastic interaction between the particles. Its mechanical parameters include the contact elastic modulus $E_u$, the Poisson's ratio $\nu_u$, the restitution coefficient $e_u$ and the friction coefficient $\mu_u$. If particles are bonded, the bond part will act in

parallel to the contact part. The bonded part is described by the bond elastic modulus $E_b$, the Poisson's ratio $\nu_b$ and the shear and tensile strength $\sigma_s$ and $\sigma_t$.

To reduce the number of variables we assume $\nu_u = \nu_b \triangleq \nu_{particle}$, $E_u = E_b \triangleq E^{particle}$ and $\sigma_s = \sigma_t \triangleq \sigma_c$. A linear relation between particle and macroscopic elastic modulus and between particle and macroscopic strength has been shown by Bobillier et al. (2020). Cundall-type numerical damping ($e_u$) was used for the weak layer particles to prevent spurious oscillations

affecting the stability of the system. However, no numerical damping was used for the slab particles because it modified the elastic wave speed in the slab which drives dynamic crack propagation (Trottet et al., 2022). The ranges in elastic modulus and strength were selected based on literature values (Mellor, 1975; Shapiro et al., 1997; Srivastava et al., 2016; Van Herwijnen et al., 2016b) and summarized in Table 1. We applied the Cundall-type numerical damping ($e_u$) for the weak layer particles to maintain the system's stability. No numerical damping was used for the slab particles to preserve slab elastic wave speed.

During the PST, the crack propagation speed was calculated based on the distance $\Delta x$ travelled by the crack tip during the time step $\Delta t$ ($c_{wl} = \Delta x / \Delta t$).

**Material point method (MPM)**

We used the 2D MPM model developed to simulate a PST experiment and study crack propagation behavior (Trottet et al., 2022). The MPM is a hybrid Lagrangian–Eulerian particles-based numerical method, where Lagrangian particles track history-

dependent variables like position, velocity, and deformation gradient, and an Eulerian grid enables the computation of spatial gradients for these quantities and is used to solve the equation of motion. The transfer of information between the grid and particles is achieved through an interpolation scheme based on quadratic B-splines and a FLIP/PIC algorithm. The slab is considered elastic and the weak layer is simulated using a modified cam clay yield surface with a hardening/softening model that allows for volumetric collapse (Gaume et al., 2018). The numerical set-up consists of a PST geometry with a bilayer

system consisting of a cohesive elastic slab ($H = 0.5$ m) and an elastoplastic weak layer ($h = 0.125$ m). The thickness of the weak layer was chosen in the higher range of values measured in full propagation PST experiments (typically between 0.02 and 0.30 m). To initiate crack propagation, a virtual saw is used to initiate a crack in the weak layer. Once the critical crack length ($a_c$) is reached, the crack starts to propagate spontaneously. We conducted 2D simulations of PSTs with lengths ranging from 25 to 140 m to ensure a steady crack propagation regime across all simulations. During propagation, the crack speed was





calculated from the difference in stress tip position at each time step ($c_{\mathrm{wl}} = \Delta x\, / \,\Delta t$). Similarly to DEM simulations in which the crack tip was defined on the basis of broken bonds, in the MPM simulations, the crack tip was defined on the basis of the onset of plasticity in the weak layer. A no-slip boundary condition was applied at the base of the weak layer. For a detailed description of the constitutive models and their parameters, we refer the reader to Gaume et al. (2018) and Trottet et al. (2022).

**Avalanche video sequences processing**

Estimating crack speeds from avalanche videos involves detecting significant changes in pixel intensity caused by snow surface movement associated with weak layer fracture. This method comprises three steps: (1) Video stabilization, (2) Tracking snow surface movement by detecting small changes in pixel intensity, and (3) Estimating crack speed using motion segmentation between consecutive video frames.

In the first step, the video was converted to grayscale, and an optical flow algorithm by Lucas and Kanade (1981) was used to
stabilize the videos. The second step utilized the principles of Eulerian video magnification (Wu et al., 2012) to detect small pixel intensity changes at the snow surface. Finally, pixel intensity is compared with the first temporal derivative to identify significant changes in pixel intensity at various locations. This technique captures the slab motion induced by crack propagation within the weak snow layer below, allowing estimation of crack propagation speed. More details about the method can be found in Simenhois et al. (2023).

**Elastic wave speed in snow**

It is known that density alone is not sufficient to describe snow mechanical properties, as, for a given density, values may scatter by orders of magnitude due to differences in snow microstructure (Mellor, 1975). Furthermore, in slab avalanches, the snow slab usually consists of several layers of different snow types. Therefore, to model the complete physics of slab avalanche release, snow temperature, strain rate, snow density and microstructure would have to be considered. As many of the processes
in snow mechanics are still poorly understood, the slab is therefore usually modelled as a uniform layer with a bulk density and an effective elastic modulus (Van Herwijnen et al., 2016a). Therefore, to calculate the velocity of elastic waves in snow, we used an approximation for the effective modulus of the slab based on the density according to the laboratory experiments of Scapozza (2004), which provides values similar to those measured in PST experiments (Van Herwijnen et al., 2016a). The following power-law relationship was used: $E = 5.07 \times 10^9 \left(\frac{\rho}{\rho_{\mathrm{ice}}}\right)^{5.13}$ with $\rho_{\mathrm{ice}} = 917\ \mathrm{kg\ m^{-3}}$. The shear wave velocity in
the slab can then be defined as $c_{\mathrm{s}}^{\mathrm{slab}} = \sqrt{G/\rho}$ with the shear modulus $G = \frac{E}{2(1+\upsilon)}$ assuming a Poisson's ratio of $\upsilon = 0.3$ (Mellor and Smith, 1966).

**Table 1: Mechanical properties used in the DEM-PST and MPM-PST simulations.**

An asterisk (*) indicates those properties that were varied within the ranges indicated in square brackets.





| Mechanical properties | DEM-PST | MPM-PST |
|---|---|---|
| Mean weak layer density (kg m$^{-3}$) | 110 | 100 |
| * Mean slab layer density (kg m$^{-3}$) | 250, [200 – 300] | 250 |
| * Slab elastic modulus $E_{slab}$ (MPa) | 10, [1 – 25] | 10, [5-30] |
| * Weak layer elastic modulus $E_{wl}$ (MPa) | 1, [0.3 – 2.3] | 1, [ 0.5 - 2] |
| * Weak layer shear strength (kPa) | 1.2, [0.59 – 1.73] | 1.1 |
| Slab porosity | 45 % | continuum |
| Weak layer porosity | 80 % | continuum |
| Weak layer height (m) | 0.02 | 0.125 |
| Weak layer collapse height (m) | 0.01 | 0.0125 |
| Slab height (m) | 0.5 | 0.5 |
| * Slope angle $\psi$ (°) | 0 – 45 | 0 – 50 |

## 3. Results

### Crack propagation regimes

We analyzed the crack propagation speed from two different numerical methods (DEM and MPM, Figure 2a,b) and from field measurements (Figure 2c). We used data from Bobillier et al. (2024) who presented 79 flat and 6 tilted DEM-PST simulations (Figure 2a). For the MPM simulations, we rely on data from Trottet et al. (2022) who described 48 flat MPM-PST simulations, 191 tilted MPM-PST simulations. The field measurements include 222 PSTs with 192 from low angle terrain with $\psi < 30°$ (Van Herwijnen et al., 2016a) and results from the analysis of avalanche videos, namely 6 cross-slope and 5 down-slope speed estimates (Figure 2b,c), which are described in detail in Trottet et al. (2022).


The numerical PSTs on flat terrain, short field PSTs, and cross-slope propagation exhibited normalized speeds (normalized by the slab shear wave speed) below one, indicating a sub-Rayleigh propagation regime (Figure 2). In contrast, tilted numerical PSTs and down-slope propagation showed normalized speeds above 1 and concentrated around 1.6 times the slab shear wave
speed (i.e intersonic speed, Figure 2). We interpret this type of propagation regime as supershear, in analogy to supershear fracture speeds observed in other materials and earthquakes (Dunham and Archuleta, 2004; Rosakis et al., 1999).

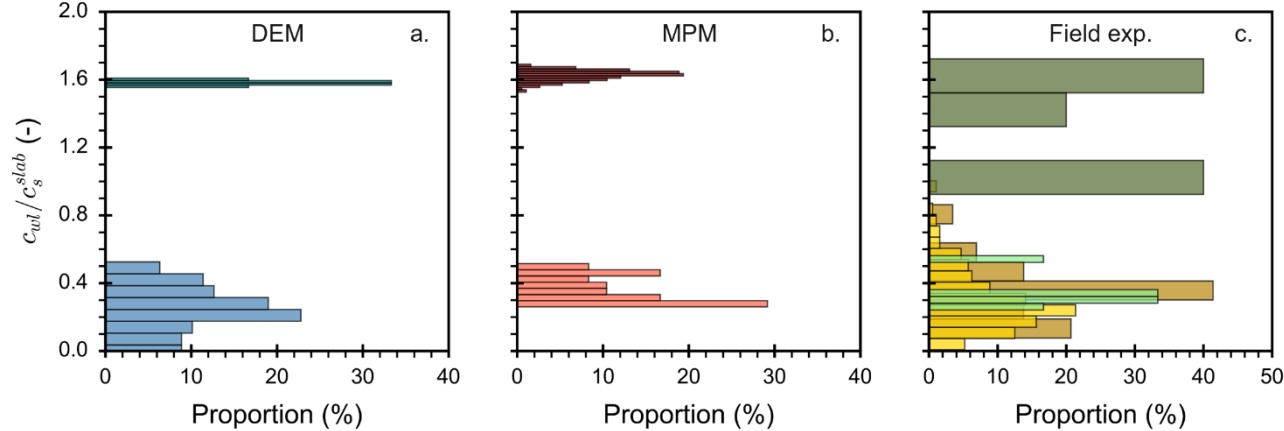

**Figure 2:** Compilation of crack propagation speeds (normalized to slab shear wave speed), relative frequency shown. (a) DEM simulations (blue $\psi = 0°$, teal $\psi = 30 - 45°$). (b) MPM simulations (orange $\psi = 0 - 20°$, maroon $\psi = 25 - 50°$). (c) Field measurements, either extracted from short PSTs (yellow $\psi < 30°$, dark yellow $\psi \geq 30°$) or derived from avalanche video sequences (light green cross-slope and dark green down-slope crack propagation speeds).

**Crack propagation dynamics and stress state**

In the numerical simulations, we investigated the dynamics of crack propagation and the associated micro-mechanical stress state along the PST column, using DEM-PST and MPM-PST simulations. The mechanical properties defined for the simulation
are summarized in Table 1. Our observations revealed two distinct crack propagation regimes, sub-Rayleigh and supershear, as illustrated in Figure 2. These regimes exhibited consistency across both numerical methods employed, with respect to the stress state, displacement fields as well as crack propagation speeds.

Figure 3a demonstrates that on flat terrain (orange and blue lines), the crack covered a shorter distance for a given time compared to the slab shear wave (black dashed line). This distinctive behavior characterizes the sub-Rayleigh propagation
regime. During this regime, the propagation of the crack was driven by a mixed-mode fracture in which compression appears to be the main driver. The stress distribution remained consistent from the initial stages of self-sustained propagation (Figure 3b) to established steady-state propagation (Figure 3c).

On inclined terrain ($\psi = 30°$) crack propagation rapidly diverged from its behavior on low angle terrain, exceeding the slab shear wave speed (Figure 3a) and transitioning to a steady-state supershear regime. During the first meters, the propagation
speed was similar for flat and inclined terrain. In the supershear regime, crack propagation was also induced by a mixed-mode





fracture within the weak layer. Nevertheless, contrary to the sub-Rayleigh regime (Figure 3b,c,d), the main stress component in the supershear regime was in shear, as shown in Figure 3e.



**Figure 3: Crack propagation dynamics and stress state for slope angles of 0° and 35° obtained with DEM-PST and MPM-PST simulations. (a) Spatio-temporal evolution of the stress tip location. Blue lines correspond to DEM simulations (blue $\psi = 0°$, teal $\psi = 30°$) and red lines to MPM simulations (orange $\psi = 0°$, maroon $\psi = 35°$). The dashed black line shows the crack tip location corresponding to the slab shear wave speed. (b-e) Normal stress $\sigma_{zz}$ (1) and shear stress $\tau_{xz}$ (2) for two slope angles and stress tip locations: 1.9 m, 17 m.**





## 4. Discussion


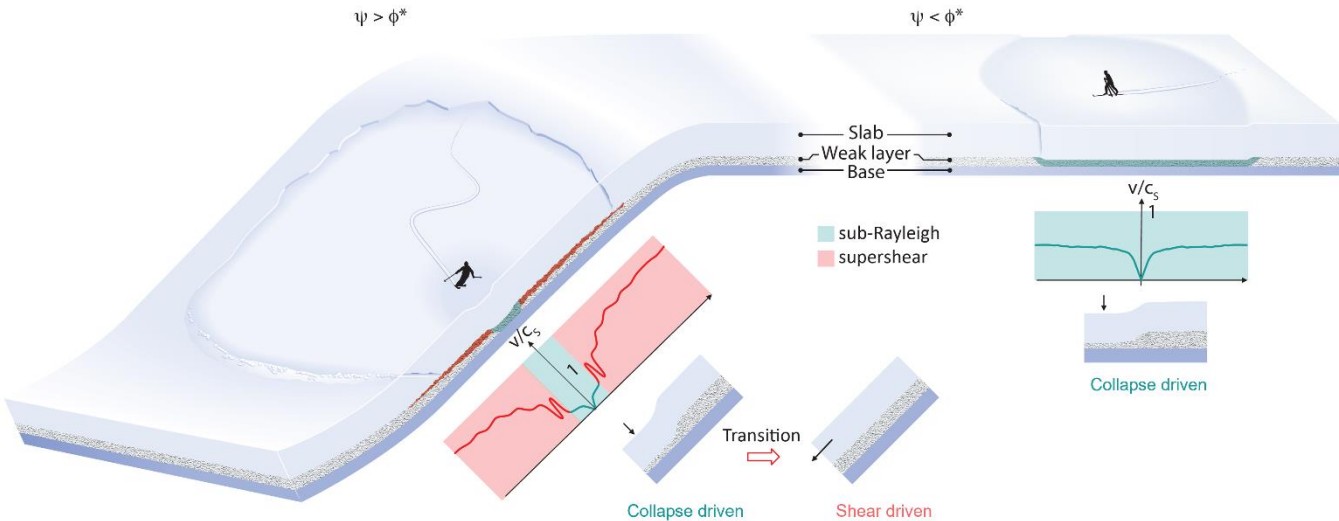

**Figure 4: Conceptual diagram of dry-snow slab avalanche initiation. On flat terrain (right), after mixed-mode failure, crack propagation in the weak layer is driven by the compressive component resulting from slab bending due to structural collapse. On a steep slope (left), crack propagation in the weak layer after a mixed-mode failure transitions from an initial propagation also driven by structural collapse to the super-shear regime, where the driving component is in shear due to the gravitational pull of the slab.**

This study highlighted the crack propagation behavior in buried weak snow layers by comparing field measurements and two distinct numerical modeling methods: DEM and MPM. Measurements and modeling offer complementary perspectives, each contributing to a multi-faceted analysis and understanding of the crack propagation regimes. While experiments provide valuable real-world data, numerical methods offer the advantage of detailed analysis of the underlying mechanism. This

interaction between experimental and numerical approaches strengthens our overall interpretation of crack propagation dynamics. With the field data, we calculated crack propagation speed using digital image correlation (DIC) and Eulerian video detection (EVD) analysis, following the procedures described by Bergfeld et al. (2022) and Simenhois et al. (2023). Through numerical simulations, we determined the crack speed based on the stress tip position, as previously suggested by Bobillier et al. (2021). These techniques allowed us to derive crack propagation speeds and gain new insights into the weak layer fracture

behavior. In addition, we investigated the dynamics of crack propagation and the associated micro-mechanical stress state along the PST column using the numerical simulations. MPM relies on continuum mechanics and macroscopic constitutive laws; in contrast, DEM simulates interactions between discrete particles with simple contact models. The good agreement between DEM and MPM results reinforces the confidence in our findings. Our compilation revealed the existence of two distinct crack propagation regimes, sub-Rayleigh and supershear.





The **sub-Rayleigh regime** is characterized by a crack propagation speed lower than the slab shear wave speed. Our observations showed that the weak layer collapse and subsequent slab bending induced a substantial concentration of compressive stresses (mode -I), leading to self-sustained crack propagation, which aligns with the stress state analysis conducted with both numerical models (Figure 4, collapse driven). This behavior emphasizes the crucial role of weak layer porosity and slab stiffness for propagation dynamics on low angle terrain. Theoretical crack propagation models such as the

solitary wave model developed by Heierli et al. (2008) included the weak layer collapse height but no other weak layer properties (Heierli et al., 2008). Siron et al. (2023) solved the problem using a Hamiltonian approach using a Timoshenko beam (internal shear forces included) and accounted for weak layer properties. In particular, Siron et al. (2023) showed the importance of the energy dissipation during the weak layer volumetric compaction on the crack speed, similar to the field PST results presented by Bergfeld et al. (2023). However, theoretical models still suffer from unknown assumptions of boundary

conditions, especially regarding the displacement gradient at the crack tip, an issue that could be resolved in the future using results from numerical models such as those presented in this paper. Based on DEM simulation results for long PST experiments, Bobillier et al. (2024) proposed a semi-empirical model to offer a crack speed prediction and a functional form for further analytical model investigation. In the present work, we report a wide range of crack propagation speeds in this sub-Rayleigh regime, typically between 0.05 and 0.8 $c_s^{slab}$ in line with the upper limit ($c_s^{slab}$) calculated by Siron et al. (2023).

On steep slopes, a **transition** was observed: the dominant fracture mode changes when the crack extends beyond the so-called super-critical crack length (approximately 5 m). Below this length, weak layer fracture is driven by the stress state observed during the sub-Rayleigh regime (Figure 4, collapse driven) where the propagation is primarily driven by the volumetric weak layer collapse, causing the slab to bend and induce stress concentrations at the crack tip. However, the bending induced stress is limited and does no longer increase after the so-called touch-down distance. On the other hand, as the crack propagates on

a steep slope, the tension in the slab continuously increases (without slab fractures) and becomes predominant over the slab bending. DEM-PST and MPM-PST models clearly show this transition to a regime where the driving stress at the crack tip is predominantly in shear. During the transition between the two regimes, both numerical methods show the apparition of a daughter crack ahead of the mother crack induced by shear stress akin to the Burridge-Andrews mechanism. For propagation distances beyond the super-critical crack length, the fracture is driven by slope-parallel gravitational forces (tension) and shear

stresses (mode II) within the weak layer designated as supershear regime (Figure 4, shear driven).

    The **supershear** regime refers to crack propagation speeds exceeding the slab shear wave speed. Earthquake ruptures exhibit similar mechanisms during supershear rupture propagation (Dunham, 2007; Andrews, 1976; Burridge, 1973). During the supershear crack propagation regime, a pure shear model, including the slab mechanical properties only, was found sufficient to approximate crack speed (Trottet et al., 2022). The observed agreement between simulation results and field observations,

enhances confidence in the accuracy of numerical predictions and their applicability to real-world avalanche scenarios. Trottet et al. (2022) suggest that the propagation mechanism driving large slab avalanches, particularly those involving hard slabs, would be in the supershear regime. Here, hard slab refers to high density, inducing rigid slab and large tensile strength and





promoting the supershear transition to occur. This finding is crucial for estimating the size of slab avalanche releases and, consequently, the avalanche hazard. Since the supershear crack propagation regime is driven by the slab properties, and the

structural collapse of the weak layer is not an essential driver, this has recently motivated the development of a depth-averaged MPM model in which the weak layer is considered a shear interface for simulating large avalanche release zones (Guillet et al., 2023). This approach (DAMPM) has been recently used to evaluate the influence of slab depth spatial variability on the avalanche release size (Meloche et al., 2024). In general, our numerical simulations and large field measurements suggest that crack speeds observed in small-scale PSTs may not be representative of slope-scale crack speeds as preliminary suggested by

Bobillier et al. (2021).

While our study represents a significant step forward, future research should include slope scale experiments and simulations to investigate the frequency of occurrence and three-dimensional propagation patterns of these crack propagation regimes. The effect of snowpack properties (e.g., slab density, slab strength) and variability on the supershear crack propagation regime should also be investigated. Furthermore, the incorporation of topographic effects and slab failure mechanisms into numerical

models will provide a better understanding of avalanche formation processes. Addressing these research gaps will contribute to the refinement of avalanche forecasting models and improve risk assessment strategies in snow-covered terrain.

## 5. Conclusion

Forecasting avalanche hazard is vital and relies on a solid understanding of avalanche-triggering processes. In this study, we investigated the crack propagation behavior in weak snow layers using field measurements and numerical methods, and we

identified two distinct crack propagation regimes: sub-Rayleigh and supershear. The sub-Rayleigh regime is characterized by a crack propagation speed lower than the slab shear wave speed, and the supershear regime refers to crack propagation speeds exceeding the slab shear wave speed. We have shown that the use of discrete (discrete element method, DEM) and continuum (material point method, MPM) numerical methods provide consistent results. These numerical models allow us to investigate the micromechanics of dynamic crack propagation. On flat terrain and for a short distance on inclined terrain, the models

highlighted the importance of the weak layer behavior for self-sustained crack propagation. For slope angles greater than the snow friction angle, we report supershear crack propagation. During this regime, the fracture is mainly driven by shear, as demonstrated based on a detailed micro-mechanical analysis of stresses. Supershear crack speed is mainly dictated by slab properties. In addition, our findings indicate that the crack propagation speed measured during small-scale experiments may not necessarily be representative of up-slope/down-slope crack speeds in slope-scale crack propagation as involved in

avalanche release.

Further research is needed to consider three-dimensional crack propagation patterns, topography effects, slab fracture, and long PST on inclined terrain. While these findings represent an essential step forward in our understanding of dry-snow slab





avalanche formation, the link between the dynamics of crack propagation and the size of the avalanche release zone remains to be established.

## 6.  Author contributions

JG, JS and AH conceived this study. GB developed the DEM model made the simulations, BT developed the MPM model made the simulations with JG. BB and AH provided the field data of the PST experiments. RS provided the analyses of the avalanche videos. All authors contributed to the interpretation of the results. GB and BT prepared the manuscript with contributions from all co-authors.

## 7.  Competing interests

The authors declare that they have no conflict of interest.

## 8.  Acknowledgments

GB was supported by a grant from the Swiss National Science Foundation (200021_16942).

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
