# Peer review of "Supershear crack propagation in snow slab avalanche release: new insights from numerical simulations and field measurements"

_Natural Hazards and Earth System Sciences, 2024_

## Author Response (AR1)

**Reply to Reviewer 1**

We would like to thank Referee 1 for reviewing our manuscript and for their helpful feedback. Below we reply to the referee's comments in **blue** color.

Forecasting the avalanche danger is vital in snow-covered mountain regions and relies on a solid comprehension of avalanche release processes. Among the various types of avalanches, accidents mainly arise from dry-snow slab avalanches. The authors used two numerical models to investigate the crack propagation speed in snow slab avalanches. They were able to explain the observed difference in crack propagation speed between field experiments and avalanches. Both models consistently showed the existence of two propagation regimes: an initial collapse-driven slower propagation is followed by a supershear crack propagation driven by shear stresses once the crack size is large enough. The combination of numerical results and experimental data confirms the model-based conclusions of previous publications.

I found the article to be well-written, clear and concise. The numerical models' findings are consistent with the experimental data, and the presented results support the conclusions.

I have only a few comments listed below.

Line 99: This sentence is a repetition of the sentence on line 95

Thank you for pointing this out. We removed the redundant sentence on line 99 to avoid repetition.

Line 151: I think you mean Figure 2c only at this place

In line 151, we refer to numerical PSTs on flat terrain, short field PSTs, and cross-slope propagation showing sub-Rayleigh propagation regime Fig 2 a,b,c. This is now clarified in the revised manuscript as follows:

The numerical PSTs on flat terrain, experimental short PSTs (flat and tilted), and field measurements of cross-slope propagation exhibited normalized speeds (normalized by the slab shear wave speed) below one, indicating a sub-Rayleigh propagation regime (Figure 2).

Figure 2: Why do you use different bin sizes, resulting in different column thickness for each dataset? Is there a specific reason or meaning?

The bin sizes in Figure 2 vary because each dataset has a different distribution and number of data points. To represent each dataset optimally, the bin widths are determined independently using Python's histogram plotting functionality with automatic bin size. This approach ensures that the gaps between bins are minimized, and the resolution of the data is preserved without over-smoothing. This is now clarified in the revised manuscript as follows:

To optimally represent each dataset, the histogram bin width was automatically determined for each case, minimizing gaps and preserving data resolution without excessive smoothing.

Figure 4: The symbols used for the crack propagation speed (v/cs) in the subplot are not consistent with the other figures. I would suggest adding an explanation in the caption since it is difficult to understand what the plots represent. Moreover, I couldn't find what  $\Phi^*$  represents. I assume that it is the threshold slope angle, but it should be specified.

Thanks for pointing out. The Figure 4 has been updated and the  $\Phi^*$  notification is now specified as follows:

Figure 4: Conceptual diagram of dry-snow slab avalanche initiation. On flat terrain (right), after mixedmode failure, crack propagation in the weak layer is driven by the compressive component resulting from slab bending due to structural collapse. On a steep slope (left), crack propagation in the weak layer after a mixed-mode failure transitions from an initial propagation also driven by structural collapse to the supershear regime, where the driving component is in shear due to the gravitational pull of the slab. The super-critical crack length only exists if the slope angle  $\psi$  is larger than the effective friction angle  $\phi^*$ .

Line 212-213: It is not clear to me what mechanism the authors are describing. Can you explain in more detail? There is no evidence in the presented data of this observation. Can you provide some reference?

This is now clarified in the revised manuscript as follows:

During the transition between the two crack propagation regimes, a secondary ("daughter") crack develops ahead of the primary ("mother") crack. The nucleation of this daughter crack is induced by shear stress concentration in front of the main crack, which is driven by compressive stress. The emergence of this secondary crack is similar to the Burridge–Andrews mechanism observed in supershear earthquakes. In our simulations, both the MPM approach (Trottet et al., 2022) and the DEM approach (Bobillier, 2022) demonstrated the onset of this Burridge–Andrews mechanism in the context of slab avalanches.

**References**

- Bobillier, G. (2022). *Micro-mechanical modeling of dynamic crack propagation in snow slab avalanche release* [Doctoral thesis, ETH Zurich]. Zurich, Switzerland. https://doi.org/10.3929/ethz-b-000588641
- Trottet, B., Simenhois, R., Bobillier, G., Bergfeld, B., van Herwijnen, A., Jiang, C. F. F., & Gaume, J. (2022). Transition from sub-Rayleigh anticrack to supershear crack propagation in snow avalanches. *Nature Physics*, 18(9), 1094–1098. https://doi.org/10.1038/s41567-022-01662-4

**Reply to Reviewer 2**

We would like to thank Referee 2 for reviewing our manuscript and for their helpful feedback. Below we reply to the three highlighted remarks in **blue** color.

The prediction of avalanche dangers relies on a solid understanding of the mechanical properties of snow slabs as well as the different structural failure processes. The paper is focused on dry-snow avalanches and the failure mechanism of crack propagation in weak layers adding an essential piece in the understanding of these failure processes based on field measurements and numerical methods. The paper is structured as follows:

- Introduction on the dynamics of crack propagation and the crack propagation dynamics from modelling and field observations
- Data and methods from snow crack propagation based on saw tests, discrete element method calculations, material point methods, video sequence processing, and the elastic wave speed in snow.
- Results on crack propagation regimes, and the crack propagation dynamics and stresses
- The discussion of results with a conceptional diagram
- Conclusions of the findings

The research results are summarized as follows:

- 1. The paper identifies two distinct crack propagation methods namely sub-Raleigh and supershear
- 2. The numerical methods discrete element method DEM and material point method MPM on these mechanisms provided consistent results.
- 3. The results show that on flat terrain the weak layer behaviour is critical for self -sustained crack propagation.
- 4. For slope angles greater than the snow friction angle the supershear crack propagation is dominant with the fraction being mainly driven by shear.
- 5. The link between the dynamics of crack propagation and size of the avalanche release zone remains to be established.

General remarks of the reviewer regarding the form of the paper and the figures:

- The paper is well written and the figures are clear
- The results from DEM and MPM align well
- The conceptual diagram is great for understanding the dominant findings on flat terrain and steep slopes

Remarks of the reviewer regarding the methods, models, and findings:

- Further investigation into the critical conditions in the transition between flat and steep terrain would be interesting as to how this affects the size of the release zone for a given topography
- Furthermore, it would be interesting to provide more insights into the snow friction angle and possible changes under different conditions over time.
- As the field tests are based on saw tests of snow columns it would be also interesting to investigate the spatial and temporal changes as this may provide further insights on the crack propagation and release areas as well as the sensitivity of the findings

In summary, the paper is well written and the findings within the scope and used methods appear to be sound. If possible, the highlighted remarks should be elaborated in the context of the reported findings prior to publication as a starting point for further understanding and research.

**Transition between flat and steep terrain**

We agree that investigating the critical conditions governing the transition between flat and steep terrain is highly relevant for understanding the potential size of avalanche release zones. Our current research focuses on the effect of slab fractures on weak layer crack propagation. Nevertheless, transitional slopes (e.g., gently rolling terrain leading into steeper sections) remain an open question. In future work, we plan to conduct further simulations across varying slope angles to identify the threshold at which the dominant failure mechanism shifts. This aspect is now introduced in the revised manuscript as follows:

Furthermore, investigating slab failure mechanisms in numerical models and incorporating topographic effects, such as slopes with increasing incline, should be a next research focus to improve our understanding of avalanche formation processes.

**Snow friction angle and its variability**

We appreciate your observation that a better understanding of the friction angle, and how it evolves under different snow conditions, is crucial. Using MPM numerical PST simulations, Trottet et al. (2022) concluded that:

"The effective friction coefficient controls the onset of the supershear transition and significantly depends on the collapse amplitude h of the weak layer. Without volumetric collapse, the effective friction angle is exactly equal to the friction angle. However, increasing collapse heights reduce the effective frictional resistance of the shear band, as reported in (van Herwijnen & Heierli, 2009). This local friction reduction enables a supershear transition for slope angles lower than the weak-layer friction angle. In effect, once the crack reaches its super-critical crack length, its sharp acceleration is associated with a significant increase of the slab section that is not supported by the weak layer, leading to unstable propagation even below the friction angle."

In DEM numerical PST simulations, for the collapse height h of the weak layer and the effective friction coefficient we assumed a constant value each. This choice was primarily due to computational power limitations. Nevertheless, we varied the slope angle.

Spatial and temporal variations in PSTs

Propagation Saw Tests (PSTs) provide a convenient method for assessing crack propagation potential in the field. However, they are sensitive to the local snowpack properties which vary in space and time. We agree with your suggestion that analyzing such spatial and temporal variability could enhance our understanding of the mechanisms discussed in this paper.

In this context, Bergfeld et al. (2023) investigated the temporal evolution of PSTs results on flat terrain. They conducted a series of 24 flat-field PST experiments, each up to 10 m long, over a 10-week period. These experiments were analyzed using digital image correlation to derive high-resolution displacement fields. The study highlighted the temporal variability in snowpack properties that influence crack arrest and full propagation. Additionally, they demonstrated that peak crack speeds were correlated with periods of high avalanche danger and natural avalanche activity.

In the future, we plan to use numerical models to simulate the effects of temporal snow properties evolution on crack propagation propensity, further refining our understanding of these dynamics. We now refer to spatial and temporal variations in the revised manuscript as follows:

Propagation Saw Tests (PSTs) provide a convenient method for assessing crack propagation potential in the field; however, they are sensitive to local snowpack properties that vary in space and time. Bergfeld et al. (2023) conducted a series of 24 flat-field PST experiments over a 10-week period and demonstrated the temporal evolution in snowpack properties influencing crack arrest and full propagation. Peak crack speeds were observed during periods of high avalanche danger and natural avalanche activity.

**References**

Bergfeld, B., van Herwijnen, A., Bobillier, G., Rosendahl, P. L., Weißgraeber, P., Adam, V., Dual, J., & Schweizer, J. (2023). Temporal evolution of crack propagation characteristics in a weak snowpack layer: conditions of crack arrest and sustained propagation. *Natural Hazards and Earth System Sciences*, 23(1), 293-315. https://doi.org/10.5194/nhess-23-293-2023 Bobillier, G. (2022). *Micro-mechanical modeling of dynamic crack propagation in snow slab avalanche release* [Doctoral thesis, ETH Zurich]. Zurich, Switzerland. https://doi.org/10.3929/ethz-b-000588641

Trottet, B., Simenhois, R., Bobillier, G., Bergfeld, B., van Herwijnen, A., Jiang, C. F. F., & Gaume, J. (2022). Transition from sub-Rayleigh anticrack to supershear crack propagation in snow avalanches. *Nature Physics*, 18(9), 1094–1098. https://doi.org/10.1038/s41567-022-01662-4

van Herwijnen, A., and Heierli, J.: Measurement of crack-face friction in collapsed weak snow layers, Geophys. Res. Lett., 36, L23502, https://doi.org/10.1029/2009GL040389, 2009.